# Inhibition of Influenza Virus Replication by Oseltamivir Derivatives

**DOI:** 10.3390/pathogens11020237

**Published:** 2022-02-11

**Authors:** Renee W. Y. Chan, Kin P. Tao, Jiqing Ye, Kevin K. Y. Lui, Xiao Yang, Cong Ma, Paul K. S. Chan

**Affiliations:** 1Department of Paediatrics, Faculty of Medicine, The Chinese University of Hong Kong, Hong Kong SAR, China; marstao@cuhk.edu.hk (K.P.T.); kwongyikevinlui@cuhk.edu.hk (K.K.Y.L.); 2Laboratory for Paediatric Respiratory Research, Li Ka Shing Institute of Health Sciences, Faculty of Medicine, The Chinese University of Hong Kong, Hong Kong SAR, China; 3CUHK-UMCU Joint Research Laboratory of Respiratory Virus & Immunobiology, The Chinese University of Hong Kong, Hong Kong SAR, China; 4Hong Kong Hub of Paediatric Excellence, The Chinese University of Hong Kong, Hong Kong SAR, China; 5Department of Applied Biology and Chemical Technology, The Hong Kong Polytechnic University, Hong Kong SAR, China; jiqing.ye@connect.polyu.hk (J.Y.); cong.ma@polyu.edu.hk (C.M.); 6Department of Microbiology, Faculty of Medicine, The Chinese University of Hong Kong, Hong Kong SAR, China; xiaoyang@cuhk.edu.hk

**Keywords:** oseltamivir derivatives, influenza virus, influenza antiviral, oseltamivir-resistant

## Abstract

Characterized by the high morbidity and mortality and seasonal surge, the influenza virus (IV) remains a major public health challenge. Oseltamivir is commonly used as a first-line antiviral. As a neuraminidase inhibitor, it attenuates the penetration of viruses through the mucus on the respiratory tract and inhibits the release of virus progeny from infected cells. However, over the years, oseltamivir-resistant strains have been detected in the IV surveillance programs. Therefore, new antivirals that circumvent the resistant strains would be of great importance. In this study, two novel secondary amine derivatives of oseltamivir CUHK326 (6f) and CUHK392 (10i), which bear heteroaryl groups of M2-S31 proton channel inhibitors, were designed, synthesized and subjected to biological evaluation using plaque assay. Influenza A virus (A/Oklahoma/447/2008, H1N1), influenza B viruses (B/HongKong/CUHK33261/2012), an oseltamivir-resistant influenza A virus (A/HongKong/CUHK71923/2009, H1N1) and an oseltamivir-resistant influenza B virus (B/HongKong/CUHK33280/2012) were included in the antiviral effect assessment compared to oseltamivir carboxylate (OC). Both novel compounds significantly reduced the plaque size of seasonal IV A and B, and performed similarly to OC at their corresponding half-maximal inhibitory concentration (IC_50_). CUHK392 (10i) functioned more effectively than CUHK326 (6f). More importantly, these compounds showed an inhibitory effect on the oseltamivir-resistant strain under 10 nM with selective index (SI) of >200.

## 1. Introduction

Despite the availability of the annually updated vaccine against influenza viruses, seasonal surge of influenza virus infection was still observed, at least before the era of Severe Acute Respiratory Syndrome Coronavirus 2 (SARS-CoV-2) pandemic. Up to date there are four antivirals, oseltamivir phosphate, zanamivir, peramivir and baloxavir marboxil, which are United States (US) Food and Drug Administration (FDA) approved and recommended by the US Centers for Disease Control and Prevention (CDC) [1]. The first three are neuraminidase (NA) inhibitors that block the viral neuraminidase enzyme, while Baloxavir is a cap-dependent endonuclease inhibitor that interferes with viral RNA transcription and blocks virus replication. These are known to be effective against both influenza A virus (IVA) and influenza B virus (IVB). There are also amantadine and rimantadine which target the M2 ion channel protein of IVA, however, they were ineffective against IVB and were associated to serious side effect. Recently adamantane drug-resistant mutants have become prevalent in most circulating A/H3N2 and some A/H1N1 viruses due to a mutation in the M2 gene. In 2013, approximately 45% of all IVA subtypes in circulation globally were resistant to the adamantanes [2]. Resistance to the adamantanes occurs rapidly within 3–5 days of use and occurs in 30–50% of both immunocompetent and immunocompromised patients therefore amantadine and rimantadine are not currently recommended as antiviral [3].

Oseltamivir is now the leading NA inhibitor that is widely used in the treatment and prevention of influenza. However, oseltamivir-resistant influenza viruses also evolved sporadically upon antiviral treatment, especially in immunocompromised patients and children [4,5,6]. During the 2008–2009 influenza season very high rates (>90%) of oseltamivir-resistant seasonal influenza, A H1N1 strains were detected [7]. In addition, nosocomial and community outbreaks of human-to-human transmission of H1N1pdm09 viruses containing H275Y mutation in NA has been reported [8,9,10]. This most frequent mutation, which confers the oseltamivir-resistant, also reduced the effectiveness of peramivir. Furthermore, some other mutations in the NA proteins of circulating viruses have been shown to affect the oseltamivir’s ability to inhibit the enzyme activity, therefore the development of novel influenza inhibitors is needed.

Phylogenetically, NAs can be divided into two groups: group 1 contains N1, N4, N5, and N8, and group 2 contains N2, N3, N6, N7, and N9 [8]. In 2006, Russell et al. disclosed that group-1 neuraminidases contain a cavity, named 150-cavity, adjacent to their active sites which suggests an additional binding pocket for drug design [9]. Crystal structures of oseltamivir in complex with NA demonstrated that the C-5-NH2 group of oseltamivir is close to the 150-cavity. The proximity indicated that C5-NH2 group could be an option for structural optimization of inhibitors binding to the 150-cavity (Appendix A). Later, a report showed that the 150-cavity of N2 NA could also be opened by NA binding to oseltamivir [3].

M2 channel blockers represent a class of traditional anti-influenza drugs [10], though their clinical use was limited by drug resistance. Recently, Wang et al. synthesized a series of M2-S31 proton channel inhibitors containing an adamantly-1-NH2+CH2-heteroaryl moiety which were effective against current viral strains resistant to M2 blockers [11,12,13]. Using a combination principle of drug design, we combined the heteroaryl groups of M2-S31 proton channel inhibitors to oseltamivir to generate a series of novel secondary amine derivatives of oseltamivir [14].

The synthesized oseltamivir derivatives were previously screened for NA inhibitory (NAI) activity. Results demonstrated that these novel derivatives inhibited the NA at a nanomolar level, in which CUHK326 (6f) and CUHK392 (10i) inhibited NA (A/H3N2) with half maximal inhibitory concentration (IC_50_) values of 1.92 ± 0.24 nM and 1.63 ± 0.16 nM, respectively. In addition, in silico predictions indicated that the absorption, distribution, metabolism and elimination (ADME) properties of the derivatives were comparable to oseltamivir [14]. This study aims to quantitate and compare the effect of these novel antiviral compounds by the classical plaque inhibition assay.

## 2. Results

### 2.1. Cytotoxicity of the Compounds

Before examining the anti-influenza virus activities, the effect of the compounds CUHK326 (6f) and CUHK392 (10i) and oseltamivir carboxylate (OC) on the cell growth and viability were tested on Madin-Darby Canine Kidney (MDCK) cells by neutral red assay. Compared with vehicle control (0.01% DMSO), CUHK326 (6f), and 320 nM for CUHK392 (10i) did not induce cytotoxicity even at 200 times of their IC_50_ determined from previous NAI assay (Figure 1A–C) as observed after 3 days of incubation. At 1000 times of their pre-determined IC_50_ (1.9 µM and 1.6 µM respectively), extensive cell death was observed. The calculated cytotoxicity concentration 50% (CC_50_) in MDCK cells for CUHK326 (6f) and CUHK392 (10i) were 1.6 µM and 1.55 µM.

A seasonal H1N1 IVA (A/Oklahoma/447/2008, IVA 447) was used as a comparator virus in this study. The inhibitory dose range of the compounds were tested by adding 100 TCID_50_ of IVA447 to the MDCK cells. Vehicle alone did not suppress the development of virus induced CPE while partial rescue of the cells was observed with OC treatment at 0.0125 nM (Figure 1D). Consistent with previous NAI findings, CUHK326 (6f) and CUHK392 (10i) exhibited an IC_50_ higher than that of OC. Partial inhibition of IVA447 was identified at the range of 0.1 to 0.4 nM while both compounds exerted complete inhibition at ≥ 0.8 nM. This indicates that addition of heteroaryl groups of M2-S31 proton channel inhibitors altered the IC_50_ of the oseltamivir against IVA infection in vitro.

### 2.2. Plaque Formation of the Tested Influenza Viruses

All tested influenza virus strains formed well-defined plaques on MDCK cells. A seasonal IVB B/Hong Kong/CUHK33261/2012 (IVB 33261), an oseltamivir resistant H1N1 pandemic IVA, A/Hong Kong/CUHK71923/2009 (ORIVA 71923), and an oseltamivir resistant IVB B/Hong Kong/CUHK33280/2012 (ORIVB 33280) were isolated from nasopharyngeal aspirates collected from patients admitted to Prince of Wales Hospital, Hong Kong. IVA447 had bigger plaques than the rest of the viruses (IVB 33261, *p* = 0.004; ORIVB 33280, *p* = 0.030; ORIVA 71923, *p* = 0.007, Table 1).

### 2.3. Inhibition of Conventional IVA and IVB by Oseltamivir Carboxylate Control

As a proof-of-concept experiment, the plaque number and size in the vehicle control were first compared with Phosphate Buffered Saline (PBS) and the treatments of OC. The plaque numbers were comparable amongst all treatments and no difference in plaque size between vehicle and PBS controls was observed.

Moreover, 0.1 nM and 0.2 nM OC treatments reduced the plaque size of IVA 447 (*p* < 0.001 and *p* < 0.001) and IVB33261 (*p* = 0.017 and *p* < 0.001), respectively, represented the validity of the experimental setting. The reduction in plaque size but not the plaque number indicates the suppression by OC, the neuraminidase inhibitor, is on the subsequent rounds of influenza virus replication but not at the initial infection.

Following the idea of oseltamivir resistance, OC had no effect on the size of ORIVB 33280 (*p* = 0.404) and ORIVA 71923 (*p* = 0.999) as in Table 1. When double dose of OC was applied, i.e., 0.2 nM, it did not alter the oseltamivir resistant phenotype, and no suppression in the plaque size of ORIVB 33280 (2.47 ± 0.17 mm^2^ to 1.82 ± 0.22 mm^2^, *p* = 0.156) and ORIVA 71923 were observed (2.40 ± 0.17 mm^2^ to 2.26 ± 0.16 mm^2^, *p* = 0.999, Table 1). The calculated IC_50_ of OC in inhibiting IVA 447 and IVB 33261 were 0.0112 nM and 0.00114 nM respectively. Previous studies showed that mutations leading to oseltamivir resistance lowered the sensitivity of OC by 200–1000 fold and 220–300 fold compared with their wildtype IAV and IBV counterparts [15]. The IC_50_ of the OC against oseltamivir-resistant strains were not determined, as double dosing of OC could only suppress the ORIVA 71923 and ORIVB 33280 by 5.57% and 20.79%, respectively.

### 2.4. Inhibition of Influenza Replication by Novel Oseltamivir-Derivatives

The novel compounds, CUHK326 (6f) and CUHK392 (10i) at their IC_50_ from the previous NAI assay (1.92 nM and 1.63 nM, respectively), [14] could significantly reduce the plaque size of IVA 447, IVB33261 and ORIVB 33280 (Table 2). In addition, CUHK392 (10i) had a better inhibition than CUHK326 (6f) to a plaque size of IVA447 (0.59 ± 0.04 mm^2^ vs. 1.38 ± 0.12 mm^2^, *p* < 0.001) at their IC_50_ and to a plaque size of IVB33261 (0.81 ± 0.11 mm^2^ vs. 1.06 ± 0.18 mm^2^, *p* = 0.026) at 2 IC_50_. Moreover, CUHK392 (10i) performed better than CUHK326 (6f) in suppressing ORIVB 33280 at both concentrations (*p* = 0.034 and *p* = 0.003, respectively). This may be explained by the electron-donating groups substituted at C-5-NH-furanyl in CUHK392 (10i) were favored for the inhibitory activity with NA by the improved H-bond interactions of acetamide. Furthermore, at double dose (2 IC_50_) of CUHK326 (6f) and CUHK392 (10i), i.e., 3.8 nM and 3.2 nM, respectively, these compounds were more potent in reducing the plaque size in all tested influenza viruses than OC at its 2 IC_50_, i.e., 0.2 nM.

### 2.5. Inhibition of Oseltamivir-Resistant IVA Isolate by Novel Oseltamivir-Derivatives

Though compounds CUHK326 (6f) and CUHK392 (10i) did not suppress ORIVA 71923 at their IC_50_ (*p* = 0.59 and *p* = 0.08, respectively), compounds provided comparable inhibitory effect to ORIVA71923 at their 2 IC_50._ CUHK326 (6f) reduced the plaque size by an average of 50% (2.40 ± 0.17 mm^2^ to 1.20 ± 0.12 mm^2^, *p* < 0.0001, Table 2) and CUHK392 (10i) reduced the plaque size by an average of 38.75% (2.40 ± 0.17 mm^2^ to 1.47 ± 0.11 mm^2^, *p* < 0.0001). Regression analysis revealed that OC has an IC_50_ lower than 0.1 nM in this MDCK platform for both IVA 447 and IVB 33261, while both CUHK326 (6f) and CUHK392 (10i) have a similar IC_50_ with that of OC in suppressing conventional strains (Figure 2 A-D. Interestingly CUHK326 (6f) and CUHK392 (10i) inhibited at least 50% inhibition at 3.8 nM and 3.2 nM, though the estimated IC_50_, i.e., 2.65 nM and 6.74 nM, by curve fitting analysis suggested a slight variation (Table 3). Based on the deduced CC_50_ values, both compounds with an SI of >200 indicated they are promising antivirals in combating oseltamivir resistance (Table 3).

## 3. Discussion

Here we validated the N-substituted oseltamivir derivatives containing C-5-NHCH2-aryl fragments of M2 channel inhibitor in suppressing IVA and IVB growth in MDCK culture. Plaque assay is a classical virology technique with its readouts in plaque number to quantify the concentration of viable viruses, and in plaque size in vitro to associate with virulence of the virus [16,17]. It has been shown that H274Y mutation in the neuraminidase gene of influenza A viruses, A/WSN/33 and A/Mississippi/3/01, was associated with reduced viral plaque areas, and confers a high level of resistance to oseltamivir without any apparent detrimental cost in terms of viral fitness and virulence in recent A(H1N1) viruses [18]. Viral growth characteristics in cell culture, including plaque morphology, have been related to pathogenesis and disease severity for several human pathogens. It was shown that small plaque variants are related to virulence and disease severity in various viruses such as herpes simplex virus, West Nile virus and respiratory syncytial virus [19,20,21]. In the case of the influenza virus, a smaller plaque size of antiviral resistance strains usually reflects a reduced fitness in vitro associated with the altered HA and NA functions [22].

Our results suggest that CUHK326 (6f) and CUHK392 (10i) did not affect the number of plaques and has a negligible role in affecting the initial binding of the virus to the cellular receptors during infection. However, these compounds effectively inhibit the subsequent rounds of IVAs and IVBs replication, including the oseltamivir-resistant strains, as indicated by the reduction in plaque size. Nevertheless, as our assessment of the viral inhibitory effect by these compounds were based on the plaque size reduction, which is not a high-throughput assay for pharmacological studies for testing drug concentration in wide range, an underestimation of their inhibitory effect to the strain with small plaque size might result. For example, an 80% plaque size reduction can be measured in a virus strain which makes big plaque (e.g., IVA447) but not for those with smaller plaque (e.g., ORIVA 71923), as an 80% plaque size reduction will make the value falls below the threshold. Further characterizations, including pharmacokinetics and the mechanism in inhibiting virus replication through combinations of inhibitors, are beneficial for better drug design of the compounds.

## 4. Conclusions

In conclusion, we synthesized a new class of influenza virus inhibitors that act by hybridizing the M2 channel inhibitor to oseltamivir and validated its effectiveness in seasonal and oseltamivir-resistant influenza A and B viruses.

## 5. Materials and Methods

### 5.1. Virus Strains

An influenza B virus (IVB) B/Hong Kong/CUHK33261/2012, an oseltamivir-resistant influenza A viruses (ORIVA) H1N1 pandemic viruses, A/Hong Kong/CUHK71923/2009, and an oseltamivir-resistant influenza B virus (ORIVB) B/Hong Kong/CUHK33280/2012 were isolated from nasopharyngeal aspirates collected from patients admitted to the Prince of Wales Hospital. The viruses were propagated in MDCK cells for no more than six passages. Seasonal H1N1 influenza A virus (IVA) (A/Oklahoma/447/2008) [23], a gift of Prof Gillian Air, University of Oklahoma Health Sciences Center, was used as the comparator virus in this study.

### 5.2. Cell Line and Virus Propagation

MDCK cells (CCL-34™, American Type Culture Collection, Manassas, MA, USA) were cultured in minimal essential medium supplemented with 10% fetal bovine serum (FBS) and 1% penicillin and streptomycin at 37 °C. MDCK cells were seeded in a T175 flask and grew until 80% confluence before viral propagation. MDCK cells were washed with PBS to remove FBS before inoculation of the virus at a multiplicity of infection (MOI) of 0.001. After 1 h of virus adsorption at 37 °C, cells were washed with PBS and replenished with MEM containing 1 µg/mL TPCK-treated trypsin (T1426, Sigma-Aldrich, Burlington, VT, USA) without FBS. The culture supernatant was harvested on day three post-infection, with 50–70% of the cells showing a cytopathic effect, by centrifugation at 4000 rpm, 4 °C for 10 min. The supernatant was harvested and stored in aliquots at −80 °C. Viral titer was then determined by plaque assay.

### 5.3. Neutral Red Cytotoxicity and Viability Assay

MDCK cells were seeded on a 96-well plate at a density of 20,000 cells per well 1 day prior to compound incubation. Cells were first washed with PBS, and replenished with 100ul medium containing compounds of respective concentrations for 72 h. Neutral Red Assay Kit-Cell Viability / Cytotoxicity kit (ab234039, Abcam, Cambridge, UK) was used according to manufacturer’s instruction. After extensive washing, cells were incubated with 150 uL Neutral Red staining solution and incubated for 4 h for color development. Optical density of 540 nm was measured by Synergy HTX Multi-Mode microplate reader (BioTek). Viability were calculated as the percentage change in absorbance reading in reference to vehicle control.

### 5.4. Plaque Assay for Virus Titration

MDCK cells were seeded on a 6-well plate at a density of 1.2 × 10^6^ per well 1 day prior to infection. Before the plaque assay, cells were washed with PBS. Influenza virus stocks or culture supernatant from the in vitro infection experiment were diluted 10-fold serially with MEM with 1% PS. 1 mL of the diluted virus/ sample was added to each of the six wells, and the plate was rocked gently every 15 min for 1 h in a 37 °C incubator with 5% CO_2_. After the virus adsorption, the inoculum was discarded and the cells were washed with PBS. In total, 2 mL agarose overlay reconstituted from MEM containing 1 µg/mL TPCK-treated trypsin in 0.8% SeaKem^®^ LE Agarose (50002, Lonza, Basel, Switzerland) was added to each well. Upon the solidification of the agarose, the assay plates were incubated upside-down in a 37 °C incubator with 5% CO_2_ for 3 days. On day three post inoculation, 10% PBS buffered formalin was added onto the agarose for cell fixation. The agarose was removed from the well and the MDCK cell was subsequently stained by 1% crystal violet (1 g crystal violet dissolved in 10 mL 95% ethanol and 90 mL MilliQ water) for 1 h. The stained monolayer was then washed and air-dried.

### 5.5. Compound Anti-Influenza Effectiveness Assessment

To examine the effectiveness of CUHK326 (6f) and CUHK392 (10i) in inhibiting influenza virus replication, we included 0.1 nM and 0.2 nM oseltamivir carboxylate (OC) as the positive control. In our previous studies [14], the IC_50_ of OC, CUHK326 (6f) and CUHK392 (10i) were 0.1 nM, 1.9 nM and 1.6 nM, respectively, as determined by a neuraminidase activity assay. Therefore, 1.9 nM and 3.8 nM of CUHK326 (6f); and 1.6 nM and 3.2 nM of CUHK392 (10i) were reconstituted using 0.01% DMSO. Briefly, each well of the MDCK cells was inoculated with 50 plaque formation units (PFU) of the virus in 1 mL inoculum at 37 °C for 1 h. After the virus adsorption, cells were washed with PBS for three times and then overlaid with 2 mL agarose containing the respective concentrations of antiviral compounds, or equal amount of PBS or DMSO as the mock-treated and vehicle control, respectively. Technical duplicates were included in each experiment, and the experiment was carried out at least three times.

### 5.6. Plaque Measurements

For virus titration, the number of plaques in each virus dilution was used to calculate the concentration of the virus stock expressed in PFU. For the compound antiviral effectiveness assessment, the total number of plaques in each well and the area of individual plaque were analyzed using OpenCFU software with a threshold value of 3 [24]. Plaques with an area of less than 25 pixels (<0.18 mm^2^ in area) were excluded from further analysis. Representative scan images of the plaque assay plates were presented in Appendix A.

### 5.7. Tissue Culture Infectivity Dose (TCID50) Based Assay

MDCK cells were seeded on 96-well tissue culture plates 1 day before the viral titration assay. Cells were washed once with PBS. Virus samples or culture supernatants were titrated in serial half-log10 dilutions with the corresponding culture medium before adding the diluted virus to the cell plates in quadruplicate. The highest viral dilution leading to CPE was recorded and the TCID50 was calculated using the Karber method. The infectivity was monitored by the infectious viral load in the supernatant, as quantitated by viral titration in MDCK cells.

To screen the effective dose of the compounds in vitro, compounds were serially diluted in a 1:2 ratio starting from 12.8 nM to 0.0125 nM using virus propagation medium, then combined with equal volume of virus containing medium with final input of 100 TCID_50_ per well. The mixtures were transferred to MDCK plates and the CPE development at respective dose was recorded after 72 h.

### 5.8. Statistical Analysis

The plaque number yielded from each drug treatment was expressed in median and interquartile range (IQR) while the percentage of plaque number of the DMSO vehicle control of the same set of experiment. The plaque size (mm^2^) was expressed in mean and standard error of means. The percentage of plaque size reduction after treatments was calculated by the average plaque area of the vehicle control of the same set of experiment; at least three sets of the independent experiment were executed. The plaque size differences among treatments were tested using one-way ANOVA analysis, followed by Bonferroni’s multiple comparisons test. Curve fitting analysis and determination of IC_50_ were performed by non-linear regression using GraphPad PRISM 9.2. Results were deemed significant when *p* < 0.05.

## 6. Patents

The compounds used in this study have been filed, Application No. US patent Application No. 62/865,347 and WIPO Application No. PCT/CN2020/097602.

## Figures and Tables

**Figure 1 pathogens-11-00237-f001:**
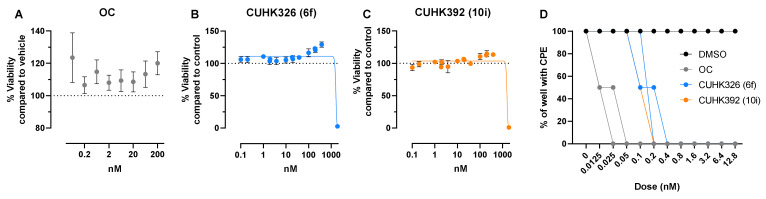
The percentage of MDCK cell viability (*n* = 4, mean ± standard deviation, SD) measured in the cytotoxicity assay of (**A**) OC, (**B**) CUHK326 (6f), and (**C**) CUHK392 (10i) revealed by neutral red assay at 3 days post incubation. The percentage of each drug concentration was in reference to vehicle control. (**D**) Percentage of well with cytopathic effect with 100 TCID_50_ IVA447 at 3 days post infection with DMSO, OC, CUHK326 (6f) and CUHK 392 (10i) by dilution assay. All MDCK wells treated with vehicle control (DMSO) had 100% CPE.

**Figure 2 pathogens-11-00237-f002:**
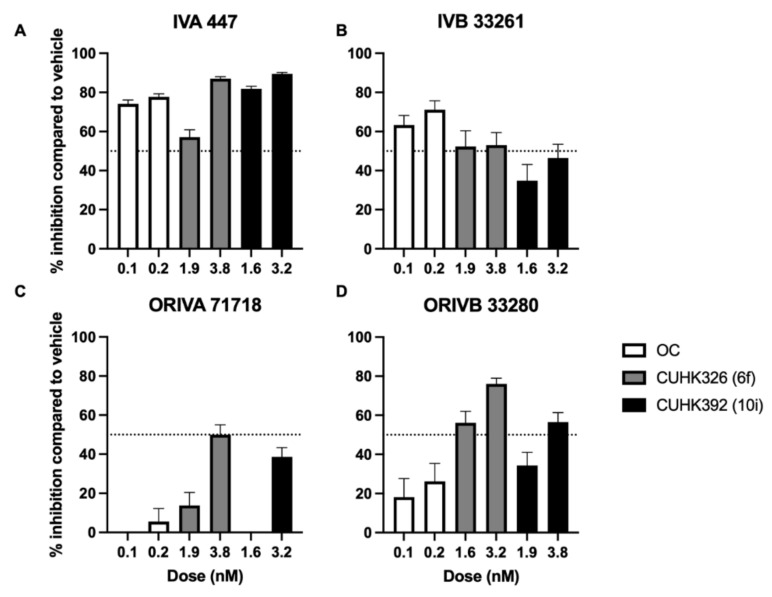
Inhibition in plaque formation of tested influenza strains in relation to vehicle control. Novel OC derivates CUHK326 (gray bars) and CUHK392 (black bars) were compared with that of OC (white bars) at their pre-determined IC50 (0.1 nM for OC, 1.9 nM for CUHK326, 1.6 nM for CUHK392) and doubled dose from previous NAI assay. (**A**) IVA 447 (**B**) IVB 33261 (**C**) ORIVA 71923 (**D**) ORIVB 33280. Error bars indicates the SEM from three independent experiments.

**Table 1 pathogens-11-00237-t001:** Plaque number (median ± IQR) and size (mean ± SEM) of IVA and IVBs in vehicle, PBS, 0.1 nM (IC_50_ from previous NAI experiment) and 0.2 nM (2 IC_50_) oseltamivir carboxylate treatments.

Viruses	Vehicle ^1^		PBS		0.1 nM OC		0.2 nM OC	
	Count ^2^	Size ^3^ (mm^2^)	Count	Size (mm^2^)	Count	Size (mm^2^)	Count	Size (mm^2^)
IVA 447	43 ± 9	3.23 ± 0.23	49 ± 11	3.40 ± 0.13	50 ± 14	0.83 ± 0.07	32 ± 10	0.72 ± 0.05
IVB 33261	29 ± 10	2.21 ± 0.19	34 ± 32	1.75 ± 0.16	26 ± 4	1.44 ± 0.19	32 ± 0	1.18 ± 0.15
ORIVB 33280	41 ± 3	2.47 ± 0.17	34 ± 40	1.77 ± 0.20	27 ± 22	2.02 ± 0.24	23 ± 16	1.82 ± 0.22
ORIVA 71923	53 ± 18	2.40 ± 0.17	52 ± 27	2.68 ± 0.12	39 ± 16	2.66 ± 0.25	49 ± 4	2.26 ± 0.16

^1^ 0.01%DMSO in MEM medium. ^2^ Frequency of plaque (median ± IQR), Kruskal–Wallis test followed by Dunn’s multiple comparison test were conducted to test for the differences in plaque count. ^3^ Size of plaque is expressed as mean ± standard error of means and the plaque size difference of the same virus between vehicle and treatments were tested by one way ANOVA followed by Bonferroni’s multiple comparisons test.

**Table 2 pathogens-11-00237-t002:** Plaque size (mean ± SEM) of IVAs and IVBs and the percentage of inhibition by the novel compounds with reference to vehicle controls.

	Viruses							
	IVA 447		IVB 33261		ORIVB 33280		ORIVA 71923	
Size (mm^2^)	Size ^1^	% ^2^	Size	%	Size	%	Size	%
Vehicle	3.23 ± 0.23		2.21 ± 0.19		2.47 ± 0.16		2.40 ± 0.17	
CUHK326 (6f) 1.9 nM 3.8 nM	1.38 ± 0.12 0.42 ± 0.03	−57.28 −87.00	1.06 ± 0.18 1.04 ± 0.14	−52.04 −52.94	1.62 ± 0.17 1.07 ± 0.12	−34.41 −56.68	2.07 ± 0.16 1.20 ± 0.12	−13.75 −50.00
CUHK392 (10f) 1.6 nM 3.2 nM	0.59 ± 0.04 0.34 ± 0.02	−81.73 −89.47	0.81 ± 0.11 0.64 ± 0.10	−63.34 −71.04	1.08 ± 0.15 0.59 ± 0.07	−56.28 −76.11	1.90 ± 0.15 1.47 ± 0.11	−20.83 −38.75

^1^ Size of plaque is expressed as mean ± standard error of means, and the plaque size difference of the same virus between vehicle and treatments were tested by one way ANOVA followed by Bonferroni’s multiple comparisons test. ^2^ The percentage of plaque size reduction was calculated by (mean treatment size − mean vehicle size)/mean vehicle size × 100%.

**Table 3 pathogens-11-00237-t003:** Calculated IC_50_ and selective index (SI) of tested compounds by plaque reduction assay. The IC_50_ and SI of OC against oseltamivir-resistant strains were not determined (N.D.) in current study.

	IVA 447		IVB 33261		ORIVB 33280		ORIVA 71923	
	IC_50_ (nM)	SI	IC_50_ (nM)	SI	IC_50_ (nM)	SI	IC_50_ (nM)	SI
OC	0.011	>1000	0.001	>1000	N.D.	N.D.	N.D.	N.D.
CUHK326 (6f)	0.104	>1000	0.068	>1000	2.650	603	1.794	891
CUHK392 (10i)	0.108	>1000	1.319	>1000	6.742	230	1.392	>1000

## Data Availability

The datasets generated are available from the corresponding author on reasonable request.

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
