# Peer review of "Inhibition of Influenza Virus Replication by Oseltamivir Derivatives"

_pathogens, 2022, doi:10.3390/pathogens11020237_

Round 1

Reviewer 1 Report

In this study, the authors evaluated the inhibitory effects of two novel OC derivates against four types of influenza virus, including oseltamivir-resistant virus. From the simple plaque measurement results presented, I believe that the two compounds could suppress viral replication in cells, however, the experiments were designed casually and have many flaws which cannot fully support the conclusions.

 One of the obvious flaws is the IC50 value determination. The IC50s involved in the antivirus assay were obtained from the enzyme activity assay the authors conducted in their previous study. Basically, the IC50 on cell line is much higher than that on enzyme. IC50 is one of the most fundamental parameters of drug evaluation in vitro, but it does not appear in this manuscript.

Line123-124, which stages of viral cycle can be inhibited by compounds should be tested by time-of-addition assay, not only by the size and number of plagues.

Line 150, why did not authors compare inhibitory effects of 6f and 10f with oseltamivir? If not, how to judge these two compounds are fair, good, or better?

Author Response

Q1: In this study, the authors evaluated the inhibitory effects of two novel OC derivates against four types of influenza virus, including oseltamivir-resistant virus. From the simple plaque measurement results presented, I believe that the two compounds could suppress viral replication in cells, however, the experiments were designed casually and have many flaws which cannot fully support the conclusions.

One of the obvious flaws is the IC50 value determination. The IC50s involved in the antivirus assay were obtained from the enzyme activity assay the authors conducted in their previous study. Basically, the IC50 on cell line is much higher than that on enzyme. IC50 is one of the most fundamental parameters of drug evaluation in vitro, but it does not appear in this manuscript.

Response: IC50 determination from this plaque reduction assay was performed accordingly and presented in line 243 to 249 and Figure 2 in the revised manuscript accordingly.

Q2:  Line123-124, which stages of viral cycle can be inhibited by compounds should be tested by time-of-addition assay, not only by the size and number of plagues.

Response: The mechanistic of CUHK326 (6f) and CUHK392 (10i) in influenza virus inhibition was not the focus of this study and we did not evaluate explicitly at which stage the viral cycle was altered. The number of plaque counts, which at least reflected the initiation of virus replication, was not alternated. We suspected that these drugs didn’t affect the initiation of virus replication. With the significant reduction of plaque size as shown in Table 1, we believed that these compounds act in a similarly as oseltamivir and prevent new virions from releasing.

Q3: Line 150, why did not authors compare inhibitory effects of 6f and 10f with oseltamivir? If not, how to judge these two compounds are fair, good, or better?

Response: Comparisons of 6f and 10i with oseltamivir is now emphasized after the revision. They showed similar IC50 and inhibition in suppressing conventional IVA and IVB. But for oseltamivir-resistant strains, only CUHK326 (6f) and CUHK392 (10i) but not OC showed viral inhibition.

Reviewer 2 Report

There is no complete inhibition of viral replication in the presence of the 2 novel OSLT derivatives as judged by a plaque reduction assay. Therefore, I propose to conduct such experiments using combinations of 6f or 10i with pimodivir or other inhibitors of viral pol or viral RNA synthesis.

Author Response

Q4: There is no complete inhibition of viral replication in the presence of the 2 novel OSLT derivatives as judged by a plaque reduction assay. Therefore, I propose to conduct such experiments using combinations of 6f or 10i with pimodivir or other inhibitors of viral pol or viral RNA synthesis.

Response: We thank the reviewer for this exciting suggestion. We agree with the reviewer that the inhibition exerted by 6f and 10i were not complete in the plaque size reduction assay. It would be wonderful to see if there would be combinational effect between 6f and 10i and also the two compounds with other inhibitors. However, as the purpose of this manuscript was to evaluate if these two compounds can inhibit the contemporary influenza viruses and oseltamivir-resistant influenza viruses before we can have further funding to evaluate the exact action of these drug, we were not able to perform further experiment as suggested. We will definitely perform follow up experiments to examine the role of 6f and 10i when new funding is available.

Reviewer 3 Report

The manuscript titled “Inhibition of influenza virus replication by oseltamivir derivatives” reports on a further biological investigation of two previously published oseltamivir derivatives (10.1016/j.ejmech.2019.111635). The authors stated that both derivatives displayed antiviral activity against different strains of influenza A and B viruses, still retaining activity against strains resistant to oseltamivir.

The identification of new anti-influenza agents is essential to fight influenza viruses, especially in order to prevent epidemics and pandemics.

However, this work shows little novelty. Herein, two compounds previously published have been tested at a single concentration against 4 different influenza virus strains. The used concentrations for both compounds were their EC50s obtained against A/H3N2 strain in the previously published work (10.1016/j.ejmech.2019.111635). In addition, both compounds work at concentrations about 16-19-fold higher than oseltamivir, thus resulting not so promising. Moreover, no MDCK cytotoxicity was evaluated. Only a sentence reporting that no cell density was affected by both compounds at IC50 or 2 IC50 concentrations.

Therefore, I believe that the authors must:

  • Measure the EC50 of both compounds against all four used influenza strains, reporting dose-response curves;
  • Evaluate the CC50 of both compounds against MDCK cells in order to establish a selectivity index (SI = CC50/EC50). Indeed, SI <10 suggests that the antiviral activity of the compound is partially due to a toxic effect to the host cell. Indeed, CC50 on MDCK cells was >100 µM (data from literature). Thus, to understand the CC50 of both compounds is essential to know their real value.

In addition, what if oseltamivir is tested at the same concentrations of 6f and 10i?

Author Response

The manuscript titled “Inhibition of influenza virus replication by oseltamivir derivatives” reports on a further biological investigation of two previously published oseltamivir derivatives (10.1016/j.ejmech.2019.111635). The authors stated that both derivatives displayed antiviral activity against different strains of influenza A and B viruses, still retaining activity against strains resistant to oseltamivir.

The identification of new anti-influenza agents is essential to fight influenza viruses, especially in order to prevent epidemics and pandemics.

However, this work shows little novelty. Herein, two compounds previously published have been tested at a single concentration against 4 different influenza virus strains. The used concentrations for both compounds were their EC50s obtained against A/H3N2 strain in the previously published work (10.1016/j.ejmech.2019.111635). In addition, both compounds work at concentrations about 16-19-fold higher than oseltamivir, thus resulting not so promising. Moreover, no MDCK cytotoxicity was evaluated. Only a sentence reporting that no cell density was affected by both compounds at IC50 or 2 IC50 concentrations.

Therefore, I believe that the authors must:

  • Measure the EC50of both compounds against all four used influenza strains, reporting dose-response curves;
  • Evaluate the CC50of both compounds against MDCK cells in order to establish a selectivity index (SI = CC50/EC50). Indeed, SI <10 suggests that the antiviral activity of the compound is partially due to a toxic effect to the host cell. Indeed, CC50 on MDCK cells was >100 µM (data from literature). Thus, to understand the CC50 of both compounds is essential to know their real value.

In addition, what if oseltamivir is tested at the same concentrations of 6f and 10i?

Response: We thank the reviewer for referring to the published study of our team, and those are enzymatic study without the validation by classical virology in cell culture. In this manuscript, we admit that the degree of novelty is lessen, however, it is the cornerstone to demonstrate its action in limiting the contemporary and oseltamivir-resistant influenza A and B in cell culture.

As suggested by the reviewer, we performed extra experiment to examine the possible cytotoxicity of these compound in vitro. No cytotoxicity was detected even at high concentration (380 nM), which is 200 times that of the IC50 determined by the previous enzyme assay. Therefore we were not able to work out the SI as suggested. The detailed data is available in line 93 to 113 and Figure 1 of the revised version. Moreover, the IC50 of the two compounds on cell culture have been determined and dose-response curves of both compounds against all four tested influenza strains were added accordingly in the revised Figure 2.

Moreover, comparing 6f and 10i to oseltamivir, 6f and 10i showed similar IC50 in inhibiting contemporary IVA and IVB. However, only 6f and 10i showed suppression to oseltamivir-resistant IVA and IVB strains.

Round 2

Reviewer 1 Report

The authors improved the manuscript quality though suppling some results and statements in the main text. However, the comparation of IC50 (or EC50) of OC and OC derivates still remained ambiguous. OC had similar IC50 with 6f and 10i (Line 165), but what were the exact IC50 values of OC against conventional IVA and IVB, and OC resistant strains? The comparation has not been involved in any tables and figures in this revised version.

Line 166, please provide more details on the results of OC activity against resistant strains, as it would be the most essential evidence to support the novelty of this study.

Moreover, as showed in Figure 2, how did authors determine IC50 correctly based on the drug activities calculated from only two or even one drug concentration? For example, two concentration dots showed up on the green line, and only one on the purple line. The data of more drug concentrations need to be conducted or added for better curve fitness.

Author Response

Q1: The authors improved the manuscript quality though suppling some results and statements in the main text. However, the comparation of IC50 (or EC50) of OC and OC derivates still remained ambiguous. OC had similar IC50 with 6f and 10i (Line 165), but what were the exact IC50 values of OC against conventional IVA and IVB, and OC resistant strains? The comparation has not been involved in any tables and figures in this revised version.

Line 166, please provide more details on the results of OC activity against resistant strains, as it would be the most essential evidence to support the novelty of this study.

A1: The IC50 of OC and derivates against all tested strains were now summarized in revised Figure 2E. The IC50 of OC in suppressing OC-resistant strains were not deduced in current study as there was studies showing that those mutants had 200 – 1000 times lowered sensitivity (with IC50 >1000 nM and >300 nM for IVA and IVB) than that of the wildtype, which are 100x higher than the tested IC50 of the compounds.  The novel derivates did not confer substantial advantages in suppression conventional IVA and IVB as their IC50 were comparable than that of OC. However, these compounds confer suppression to tested ORIVA and ORIVB with estimated IC50 under 10 nM.

Q2: Moreover, as showed in Figure 2, how did authors determine IC50 correctly based on the drug activities calculated from only two or even one drug concentration? For example, two concentration dots showed up on the green line, and only one on the purple line. The data of more drug concentrations need to be conducted or added for better curve fitness.

A2: Due to the time limitation and low throughput of plaque reduction assay, we are not able to expand our study with more concentrations of the compounds to be used for curve fitting purpose. We agreed that estimation of IC50 with just 3 concentrations is suboptimal. The percentage change in the reduction of plaque size at pre-determined IC50 and their doubled dose in comparison to vehicle control were now emphasized. We will increase the extensiveness on the pharmacokinetics studies if new funding is available.

Reviewer 2 Report

The study lacks novelty: "Both novel compounds performed similarly to OC at their corresponding half-maximal inhibitory concentration (IC50)".  

The study can be improved by finding drug combinations, which would overperformed OC.

Moreover, CC50, SI and selectivity indexes would be important to calculate and present.

Author Response

Q3: The study lacks novelty: "Both novel compounds performed similarly to OC at their corresponding half-maximal inhibitory concentration (IC50)". 

The study can be improved by finding drug combinations, which would overperformed OC.

Moreover, CC50, SI and selectivity indexes would be important to calculate and present.

A3: We explored the cytotoxicity and inhibition of oseltamivir-resistance strains of previously characterized OC derivates in vitro. The CC50 of both compounds are 1600 and 1555 nM, with SI of >200 and >800 for 6f and 10i respectively, which is now summarized in revised Figure 2. This indicating that low working dose of IC50 (<10nM for OC resistant strains), together with the low cytotoxicity of the compounds may be useful for combating oseltamivir-resistance strains. As mentioned in the previous rebuttal, with our current limited research resources, we are not able to perform additional drug combinations study in the current submission. It has been listed as one of the limitations of the current study.

Reviewer 3 Report

The authors satisfied most of the reviewer requests. However, EC50 values of derivatives 6f and 10i on oseltamivir-resistant strains highlight that both strains are significantly resistant to both compounds. Indeed, comparing EC50 values of both compounds against “wild-type” strains and against oseltamivir-resistant strains is evident the wide increase of EC50 values when compounds were tested against resistant strains (>10 fold). Anyway, considering the high CC50 values, both compounds still exhibit promising selectivity indexes.

Nevertheless, at this point, a comparison of the activity of both compounds with the activity of oseltamivir is essential. Accordingly, in my opinion, the authors must add EC50 values of oseltamivir against all used influenza strains and then, discuss the difference in the activity between oseltamivir and new derivatives against influenza strains.

In addition, in Figure 2A and 2B, both compounds did not reach the 100% growth inhibition of influenza strains at the used concentrations. Maybe the authors should use higher concentrations in order to obtain a correct dose-response curve. Indeed, in my opinion, the “curve” of 10i against ORIVA91923 is not acceptable for publication.

Author Response

Q4: The authors satisfied most of the reviewer requests. However, EC50 values of derivatives 6f and 10i on oseltamivir-resistant strains highlight that both strains are significantly resistant to both compounds. Indeed, comparing EC50 values of both compounds against “wild-type” strains and against oseltamivir-resistant strains is evident the wide increase of EC50 values when compounds were tested against resistant strains (>10 fold). Anyway, considering the high CC50 values, both compounds still exhibit promising selectivity indexes.

Nevertheless, at this point, a comparison of the activity of both compounds with the activity of oseltamivir is essential. Accordingly, in my opinion, the authors must add EC50 values of oseltamivir against all used influenza strains and then, discuss the difference in the activity between oseltamivir and new derivatives against influenza strains.

A4: We thank the reviewer for appreciating the extra work done in our last revision. The IC50 of OC and derivates against all tested influenza strains can be found in the revised Figure 2E. We agree that the novel derivates did not confer substantial advantages in protecting conventional IVA and IVB as their IC50 were not lowered than that of original OC. However, the two novel compounds confer suppression to the ORIVA and ORIVB with an estimated IC50 below 10 nM.

Moreover, the IC50 of OC in suppressing OC-resistant strains could not deduced in current study as there were studies showing that those mutants had 200 – 1000 times lowered sensitivity (with IC50 above 1000 nM and 300 nM for IVA and IVB) than that of the wildtype, which are at least 100x higher than the tested IC50 of the compounds. 

Q5: In addition, in Figure 2A and 2B, both compounds did not reach the 100% growth inhibition of influenza strains at the used concentrations. Maybe the authors should use higher concentrations in order to obtain a correct dose-response curve. Indeed, in my opinion, the “curve” of 10i against ORIVA91923 is not acceptable for publication.

A5: Due to the time limitation and low throughput of plaque reduction assay, we are not able to expand our study with more concentrations of the compounds to be used for curve fitting purpose. We agreed that dose-response relationship could not be determined with only 3 drug concentrations, higher and more concentrations should be tested to strengthen the IC50 estimation. The percentage change in the reduction of plaque size at pre-determined IC50 and their doubled dose in comparison to vehicle control were now used instead of exaggerated using dose-response curve. We will increase the extensiveness on the pharmacokinetics studies of both compounds if new funding is available, and the limitations of the current studies is now addressed in the discussion part.

Round 3

Reviewer 1 Report

Now I agree to publish this study on Pathogens in present version.

Reviewer 2 Report

The authors stated that they have limited resources and are therefore unable to conduct the requested experiments.